# Microstructural and Texture Evolution in Pure Niobium during Severe Plastic Deformation by Differential Speed Rolling

**DOI:** 10.3390/ma15030752

**Published:** 2022-01-19

**Authors:** Sang Yong Park, Woo Jin Kim

**Affiliations:** 1Advanced Fusion Process R&D Group, Korea Institute of Industrial Technology, Incheon 21999, Korea; persona8302@kitech.re.kr; 2Department of Materials Science and Engineering, Hongik University, Seoul 04066, Korea

**Keywords:** niobium, high ratio differential rolling (HRDSR), continuous dynamic recrystallization (CDRX), texture, severe plastic deformation

## Abstract

The evolution of the microstructure and texture in body-centered cubic (BCC) niobium (Nb) during conventional rolling and high-ratio differential rolling (HRDSR) at room temperature were compared. More effective grain refinement of the initial microstructure through continuous dynamic recrystallization (CDRX) occurred in the samples processed by HRDSR, but the overall degree of grain refinement was small, despite having undergone severe plastic deformation due to the low rate of CDRX. CDRX more preferentially proceeded on {111}<uvw> γ-fiber grains than on {001}<110> α-fiber grains. The HRDSR-processed samples exhibited weaker α-fiber and stronger γ-fiber than the conventionally processed samples, which indicates that the high shear deformation induced by HRDSR discourages the development of α-fiber while promoting the development of γ-fiber. The HRDSR processed Nb showed a high tensile strength of 450 MPa, and the major strengthening mechanism for the HRDSR-processed Nb was dislocation-density strengthening at large thickness reductions.

## 1. Introduction

Niobium (Nb) has a body-cubic centered (BCC) structure and a very high melting point (2741 K). Niobium is scientifically and practically important, since it is the main element of Nb_3_Sn- and NbTi-based superconductors [1,2]. Pure Nb has a very low strength, so its application is limited. Therefore, the strengthening of pure Nb through microstructural refinement is of interest because high-strength Nb can be applied to fabricate special superconducting components where alloying should be avoided [3].

Cold rolling has been used to refine the microstructure of pure Nb [4,5]. Sandim et al. [5] applied cold rolling (80% thickness reduction) to coarse-grained Nb and observed the deformation-driven subdivision of grains, but the obtained microstructure was quite heterogeneous and remained coarse. Abreu et al. [6] applied cold rolling to Nb polycrystalline sheets, achieving up to a 90% thickness reduction. {100}<110> (α-fiber) was developed during the rolling process, and intensities of the α-fiber components increased with the amount of cold deformation. Meanwhile, {111}<uvw> (γ-fiber) was weaker and did not increase with deformation. Severe plastic deformation (SPD) has also been applied to pure Nb. Zhu et al. [7] applied equal-channel angular pressing (ECAP) to pure-Nb single crystals with different orientations to examine the microstructural and texture evolution during a single ECAP pass and linked the obtained microstructure with the initial crystal orientation. Bernardi et al. [3] applied ECAP to [211]-oriented niobium single crystals at room temperature (using eight ECAP passes). A cube texture developed during the ECAP process. The microstructure consisted of fine crystallites arranged in a lamellar structure with sizes of ~1 µm. Popov et al. [8] investigated the evolution of the microstructure of polycrystalline Nb subjected to ECAP and high-pressure torsion (HPT), and reported a considerable reduction of grain size (to submicron size) after many ECAP passes and HPT turns.

Although the variation of microstructure and texture in pure Nb after ECAP or cold rolling has been studied as reviewed above, there are few reports on the effect of differential speed rolling (DSR) on the microstructure and texture development in pure Nb. A DSR technique has been used to fabricate the sheet materials with refined microstructure and controlled texture and has attracted many researchers and industries [9,10,11,12,13]. High-ratio DSR (HRDSR) is a special DSR technique that can produce ultrafine grains by inducing high shear straining during rolling process by adopting a large roll speed difference between upper and lower rolls [14,15,16,17,18]. This study aims to examine the effect of SPD via HRDSR on the microstructure, texture, and mechanical properties (tensile properties) of pure Nb and determine the relationship between microstructural parameters and strength.

## 2. Materials and Methods

The material in this study was a polycrystalline commercially pure Nb sheet (99.9% in purity) 20 mm wide and 2 mm thick. Rolling was conducted using a custom-made rolling machine, where the speed of the upper and lower rolls could be controlled independently [19]. The upper and lower rolls had diameters of 300 mm. In equal-speed rolling (ESR), the ratio of upper to lower roll speed was 1 (6 rpm. vs. 6 rpm); in HRDSR, the ratio was 3 (6 rpm vs. 2 rpm). In both ESR and HRDSR, the Nb sheet was rolled from 2 mm to 0.4 mm (corresponding to an 80% thickness reduction in total) through multiple passes (5–6 in total, with a thickness reduction of 10–20% per pass) at room temperature. Rolling was performed under nonlubricated condition, and no skidding was observed.

The as-received and rolled Nb sheets were sectioned along the longitudinal section (normal direction (ND)–rolling direction (RD) plane) for microstructural observation. To characterize the microstructure of the Nb sheets, a field emission–scanning electron microscope (FE-SEM; FEI, QUANTA FEG 250, FEI Company, Hillsboro, OR, USA) equipped with electron back scattering diffraction (EBSD; EDAX. Digiview, AMETEK, Inc., Berwyn, PA, USA) system was used. To prepare the samples, the surface of the samples was mechanically ground using SiC paper and polished using a 1-μm diamond suspension. The EBSD scanning step size was 1.25 µm, and the scanned area was 40 µm × 40 µm. For each sample, EBSD scans were performed four times in different areas along the middle of its longitudinal section, where the most distinguishable difference in microstructure was expected to be observed between the samples processed by ESR and HRDSR [19]. The data obtained from the EBSD scans were processed using TSL-OIM analysis software (OIM Analysis V8, AMETEK Inc., Newark, DE, USA). The grain orientation spread (GOS) method was used to determine the fractions and sizes of the dynamically recrystallized (DRXed) grains. The grains with a GOS value of ≤2.5° were considered to be DRXed grains. The density of geometrically necessary dislocations was calculated using the kernel average misorientation obtained from the EBSD analysis.

The evolution of the microtexture was analyzed by calculating the orientation distribution functions (ODFs) from the EBSD measurements. The harmonic series method, which assumes triclinic sample symmetry and superimposes a Gaussian peak for each individual orientation with a spread of 5° around the exact orientation, was applied upon calculation of texture in the Euler angle space (φ_1_ = 0–90°, Φ = 0–90°, φ_2_ = 0–90°).

Dog-bone-shaped tensile samples with a gauge length of 5 mm parallel to the RD were prepared from the rolled samples. Tensile tests were performed at 5 × 10^−4^ s^−1^ using a universal testing machine (UNITECH-M, R&B, Daejeon, Korea).

## 3. Results and Discussion

### 3.1. Initial Microstructure and Texture

Figure 1a shows the inverse pole figure (IPF) map for the microstructure of the as-received Nb sheet. The microstructure is composed of equiaxed recrystallized grains with an average size of 57.1 µm. The texture of the as-received Nb sheet was analyzed using ODFs at φ_2_ = 0° and 45° planes, and the result is shown in Figure 1b. A schematic infographic of the position of the most common BCC-rolled texture components is presented in Figure 2. The ODF maps of the as-received Nb sheet show the presence of strong {001}<110> and {001}<111> texture components.

### 3.2. Microstructure Evolution during Rolling

Figure 3a–e shows the optical micrographs of the as-received and HRDSR-processed samples at different thickness reduction ratios. With increasing thickness reduction, the microstructure deformed more severely. The deformed microstructures show both elongated and equiaxed grains at the thickness reductions of 20–60%. However, the entire microstructure became homogeneous at a thickness reduction of 80%, composed of a number of highly squeezed grains. Figure 4a–h shows the IPF maps and grain boundary (GB) maps of the ESR-processed and HRDSR-processed samples at different thickness reduction ratios. For both ESR- and HRDSR-processed samples, as the thickness reduction increased, the grains became more elongated along the rolling direction. Fine grains with sizes of 2–3 µm (indicated by arrows in Figure 4h) were only clearly observed in the sample processed by HRDSR after deformation with thickness reduction of 80%.

Figure 5a–h shows the grain size distribution plots of the samples processed by ESR and HRDSR with different thickness reduction ratios. For both materials, when the thickness reduction increased, the fraction of small grains increased. The most drastic difference between the two materials was observed at the 80% thickness reduction, where the fraction of small grains with sizes less than 5 µm was higher (27.5% vs. 15.9%), and the degree of grain size distribution was more uniform in the HRDSR-processed sample. These observations indicate that HRDSR is more effective for the grain refinement of pure Nb than conventional rolling.

Figure 6 shows the average grain sizes and aspect ratio of grains of the ESR- and HRDSR-processed samples with different thickness reduction ratios. When the ratio increased, the average grain size decreased. The HRDSR-processed samples had smaller average grain sizes than the ESR-processed samples at all thickness reduction ratios. For example, the grain sizes of the former and latter at the 80% thickness reduction were 20.6 and 29 µm, respectively. The aspect ratio of grains also decreased with increasing thickness reduction, which implies that the grain shape became more elongated when the accumulated strain increased. However, the HRDSR-processed samples exhibited a higher aspect ratio than the ESR-processed samples, indicating that elongated grains were more effectively refined via grain subdivision during the HRDSR process.

Figure 7a shows the fractions of high-, intermediate- and low-angle grain boundaries of pure Nb processed by ESR and HRDSR at different stages of rolling. For all deformed samples, low-angle grain boundaries (LAGBs) are dominant, but the fraction of high-angle grain boundaries (HAGBs) is higher in the samples processed by HRDSR. For both the ESR- and HRDSR-processed materials, after the deformation of a 20% thickness reduction, the fraction of HAGBs greatly decreased due to the development of a deformed structure, but increased at a thickness reduction above 20 or 40%. The rate of increase of the HAGB fraction with a thickness reduction above 40% was higher in the material processed by HRDSR. Figure 7b shows the fraction of dynamically recrystallized (DRXed) grains of the ESR- and HRDSR-processed samples. After rolling by 20% in thickness reduction, the fraction of DRXed grains sharply decreased to 0.7%, but increased with a further increase in thickness reduction ratio in the ESR-processed sample. At a thickness reduction of 80%, it was 4.5%. For the HRDSR-processed sample, the fraction of DRXed grains increased to 8.4% at a thickness reduction of 80%.

### 3.3. Texture Evolution during Rolling

Figure 8 shows the textures of the ESR-processed and HRDSR-processed samples obtained at different thickness reduction ratios. After rolling by ESR and HRDSR, the dominant texture ({001}<110>) remained the same, but the {112}<111> texture component disappeared after the 20% thickness reduction. To better visualize the texture features, the main BCC fibers (α- and γ-fibers) were calculated and plotted in Figure 9a,b. The ESR-processed sample exhibited a stronger α-fiber than the HRDSR-processed sample. During ESR, the intensity of the rotated cube texture increased up to a thickness reduction of 40%, and subsequently decreased at 80%. During HRDSR, the intensity of rotated cube texture also increased with thickness reduction (up to 60%) and then decreased at 80%. However, the development of the rotated cube texture during deformation was less intensive in HRDSR. During both ESR and HRDSR, the rotated cube texture intensity peak tended to move from {001}<110> to {112}<110> with increasing thickness reduction ratio. Another texture component ({111}<110>) belonging to α-fiber, whose intensity was relatively weak compared to the rotated cube texture, also increased and then decreased with increasing thickness reduction in both ESR and HRDSR. Abreu et al. [6] studied the crystallographic macrotexture of pure niobium cold rolled to 30–90% thickness reduction by X-ray diffraction and found that the α-fiber was strengthened with deformation and the (001)[110] component was dominant. These results are generally consistent with the current observations. In the γ-fiber, the ODF intensity of the sample processed by ESR increased at the thickness reduction of 20%, but decreased with a further increase in thickness reduction. The γ-fiber almost disappeared at the large thickness reductions of 60 and 80% in ESR. This result agrees with Abreu et al. [6]’s report that cold-rolled niobium did not show an increase in γ-fiber with deformation. Meanwhile, the γ-fiber intensity of the sample processed by HRDSR became more intense with increasing thickness reduction. The intensity peak tended to move from {111}<110> to {111}<112> with increasing thickness reduction. This observed γ-fiber intensity strengthening indicates that shear stress induced during HRDSR promoted the development of γ-fiber. The development of the ND//{111} texture can be beneficial, because it is known to increase the drawing ability of BCC metals [20]. Figure 2, Figure 8 and Figure 9 show that the starting material and ESR- and HRDSR-processed samples have a complete absence of the {011} texture (B and Goss shear components). This observation is consistent with the report of the absence of the {011} texture in the ECAP-processed rolled Nb sheets [21].

Figure 10a–d shows the representative misorientation profiles along arrows (1)–(8) on the IPF maps for the ESR- and HRDSR-processed Nb after thickness reductions of 20% and 80% in Figure 4. For both the ESR- and HRDSR-processed samples, the accumulated misorientation (point-to-origin) progressively increases when the position moves from the grain interior to the grain boundaries, and the maximum point-to-point and point-to-origin misorientation angles tend to increase with increasing thickness reduction ratio. However, both point-to-point and point-to-origin misorientation angles are higher in the HRDSR-processed samples. These observations indicate that continuous DRX (CDRX) occurs during ESR and HRDSR, but the CDRX rate is higher in HRDSR. The CDRX mechanism features progressive rotation of subgrains to form new (recrystallized) grains, and misorientation angles above 12–15° often indicate the accumulation of a high stored energy for occurrence of CDRX in subgrain structures [22,23]. CDRX typically occurs in metals with medium and high stacking fault energy (SFE) materials [22] such as Al (166 mJ/m^2^ [24]), ferritic steel (180 mJ/m^2^ [25]), and Nb (200 mJ/m^2^ [26]), because when the stacking fault energy is high, the recovery rate is high, so discontinuous DRX is suppressed. It is noted that the rate of CDRX in the HRDSR-processed Nb is significantly lower than that in the HRDSR-processed copper (with SFE = 78 mJ/m^2^ [27]) studied in the previous work [18], which was cold rolled by HRDSR under similar experimental conditions. The reason for this may be that the rate of dislocation annihilation by recovery is much higher in Nb, which has a much higher stacking fault energy than Cu, and thus it is more difficult to reach the critical dislocation density for forming new HAGBs by CDRX in Nb. Another important observation from the misorientation profiles in Figure 10 is that the rate of CDRX is higher in the {111}<uvw> γ-fiber grains than in the {001}<110> α-fiber grains. It has been reported [28,29] that the {001}<110> rotated cube component in BCC metals has small in-grain orientation gradients, even after large plastic deformation, which implies that the multiplication rate of dislocations is low during deformation, whereas the {111}<uvw> γ-fiber has a large strain gradient and high dislocation density due to the activation of the {110}<111> slip. For this reason, the stored energy is low for the nucleation and evolution of new grain boundaries in the {001}<110> grains, while it is high in the γ-fiber grains. This explains why recrystallized grains are preferentially observed on the {111}<uvw> γ-fiber grains (Figure 4h).

### 3.4. Tensile Properties

Figure 11a,b show the engineering stress–engineering strain curves of ESR- and HRDSR-processed samples at different thickness reduction ratios. Figure 11c,d show the changes in work hardening rate (θ = dσ/dε) and true stress (σ) as a function of true strain (ε) for the ESR- and HRDSR-processed samples, which were plotted to determine the amount of uniform elongation and the work hardening rates. The results for the yield stress, ultimate tensile strength, and tensile elongation of the ESR- and HRDSR-processed samples are plotted in Figure 11e,f, respectively. Compared to the as-received sample, the yield and ultimate tensile strength of the ESR- and HRDSR-processed samples tend to increase with increasing thickness reduction ratio. The ultimate tensile strength of the sample processed by HRDSR after the 80% thickness reduction is as high as 447.1 MPa, which is approximately 100 MPa higher than that of the ESR-processed sample at the same thickness reduction. The tensile elongation (uniform elongation) tends to rapidly decrease with increasing thickness reduction in both ESR- and HRDSR-processed samples, which is due to a significant decrease of work hardening ability after heavy plastic deformation.

The yield strength (σ_y_) of a single-phase pure Nb after deformation is expected to be principally determined by grain size and dislocation density. The strengthening by grain size reduction (∆σ_g_) can be estimated using the Hall–Petch (H-P) relation [30,31]:∆σ_g_ = *Kd*^−1/2^(1)
where *K* is the H–P constant (276 MPa × μm^1/2^ fore pure Nb [32]), and *d* is the grain size.

The strengthening by dislocations (∆σ_disl_) is typically expressed as follows [33]:∆σ_disl_ = *M**αGbρ*^1/2^(2)
where *M* (2.75 for typical BCC metals [34]) is the Taylor factor, *α* (0.47 for pure Nb [35]) is the constant, *G* (37.5 GP for pure Nb [36]) is the shear modulus, *b* (0.286 nm for pure Nb [37]) is the Burgers vector, and *ρ* is the dislocation density. In this work, we used the density values of geometrically necessary dislocations obtained from the EBSD analysis (0.59 ×, 0.98 ×, 1.52 × and 1.77 × 10^14^ m^−2^ at 20, 40, 60 and 80% thickness reduction ratios, respectively) to calculate ∆σ_disl_.

When these two contributions to strength are considered, σ_y_ can be calculated using Equation (3):σ_y_ = σ_0_ + ∆σ_g_ + ∆σ_disl_(3)
where σ_0_ is the friction stress (69 MPa for pure Nb [32]). Figure 12 shows the calculated results for the HRDSR-processed samples. The calculated values are reasonably close to the experimental data and follow the trend that the experimental data show. The analysis results conclude that as the thickness reduction increases, the contribution of dislocation strengthening to total strength increases and dislocation strengthening becomes the major strengthening mechanism at large thickness reductions.

## 4. Conclusions

Pure Nb was subjected to high-ratio differential rolling that produced severe plastic deformation during rolling. The changes in the microstructure, texture and tensile properties after rolling were examined. The following results were obtained.

1.More effective grain refinement by CDRX occurred in the samples processed by HRDSR compared to those processed by ESR, but the degree of grain refinement was low even in the HRDSR-processed samples most likely due to a very high SFE of Nb.2.The ESR-processed samples exhibited a stronger α-fiber than the HRDSR-processed samples, while the HRDSR-processed samples exhibited a stronger γ-fiber, indicating that strong shear deformation induced by HRDSR promoted the development of γ-fiber.3.CDRX preferentially occurred on the {111}<uvw> γ-fiber grains, which was attributed to the accumulation of a higher dislocation density (leading to a high strain gradient) compared to the {001}<110> rotated cube component.4.The HRDSR-processed Nb showed higher strength than the ERS-processed Nb. The strengthening mechanism for the HRDSR-processed Nb was analyzed, and dislocation strengthening was found to be the major strength mechanism.

## Figures and Tables

**Figure 1 materials-15-00752-f001:**
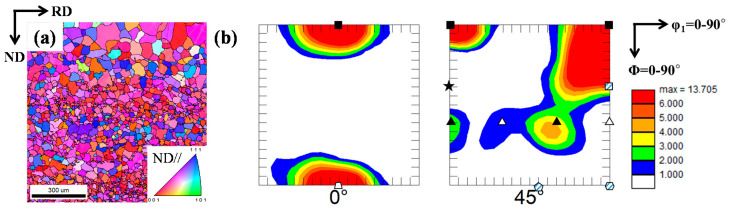
(**a**) IPF map and (**b**) ODF sections with φ_2_ = 0° and 45° of the as-received Nb sheet.

**Figure 2 materials-15-00752-f002:**
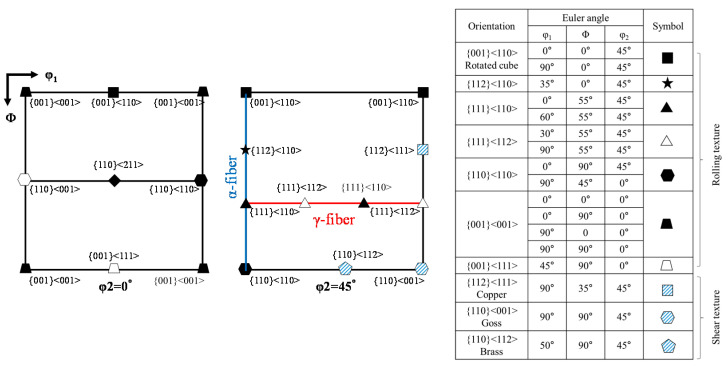
Schematic illustration of common texture components in BCC metal: ODF sections with (**a**) φ_2_ = 0° and (**b**) 45°.

**Figure 3 materials-15-00752-f003:**
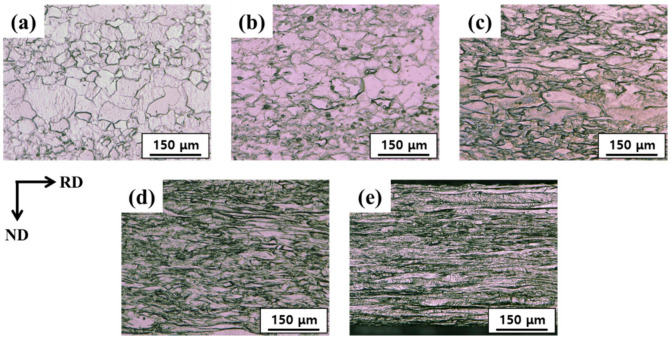
Optical microstructures of the HRDSR-processed samples with different thickness reduction ratios: (**a**) as-received, (**b**) 20%, (**c**) 40%, (**d**) 60%, and (**e**) 80%.

**Figure 4 materials-15-00752-f004:**
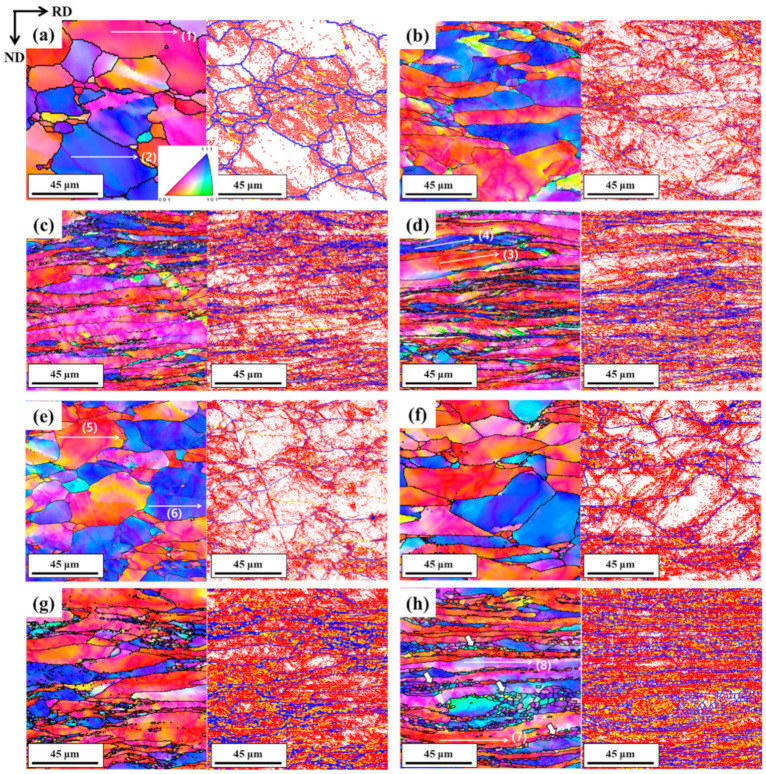
IPF maps and GB maps of the (**a**–**d**) ESR- and (**e**,**f**) HRDSR-processed samples at different thickness reduction ratios of (**a**,**e**) 20%, (**b**,**f**) 40%, (**c**,**g**) 60% and (**d**,**h**) 80%. In the GB maps, low-angle grain boundaries (2–5°) are shown in red, intermediate-angle grain boundaries (5–15°) are in yellow, and high-angle grain boundaries (>15°) are in blue.

**Figure 5 materials-15-00752-f005:**
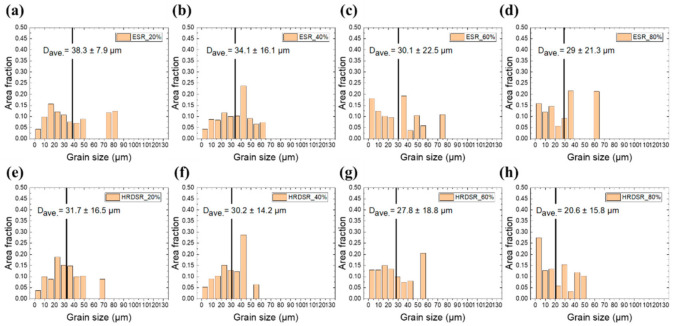
Grain size distribution plots for the ESR-processed samples with different thickness reduction ratios: (**a**) 20%, (**b**) 40%, (**c**) 60%, and (**d**) 80%. Grain size distribution plots for the HRDSR-processed samples with different thickness reduction ratios: (**e**) 20%, (**f**) 40%, (**g**) 60%, and (**h**) 80%.

**Figure 6 materials-15-00752-f006:**
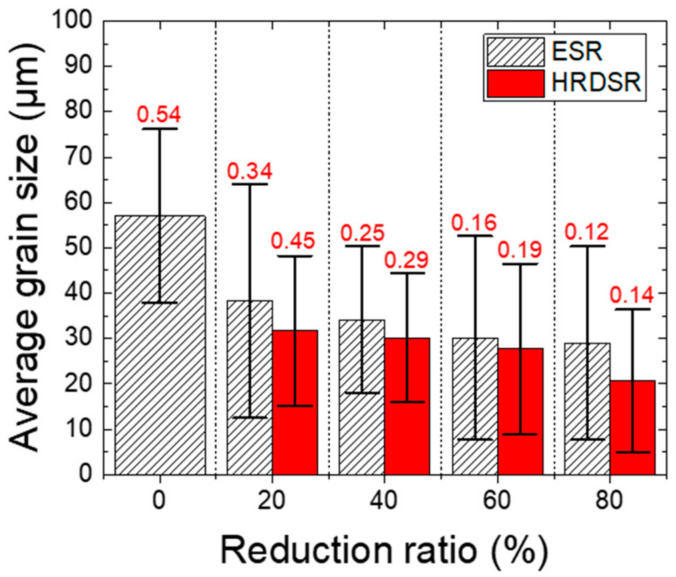
Average grain sizes (with standard deviation) and aspect ratio of grains (marked by red color numbers) of the ESR- and HRDSR-processed samples with different thickness reduction ratios calculated from the EBSD data.

**Figure 7 materials-15-00752-f007:**
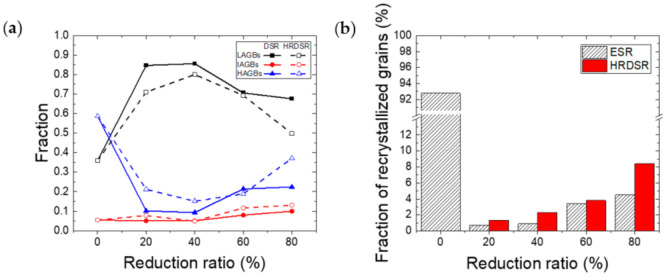
(**a**) Fractions of high-, intermediate- and low-angle grain boundaries of pure Nb processed by ESR and HRDSR at different stages of rolling. (**b**) Fraction of DRXed grains of the ESR- and HRDSR-processed samples.

**Figure 8 materials-15-00752-f008:**
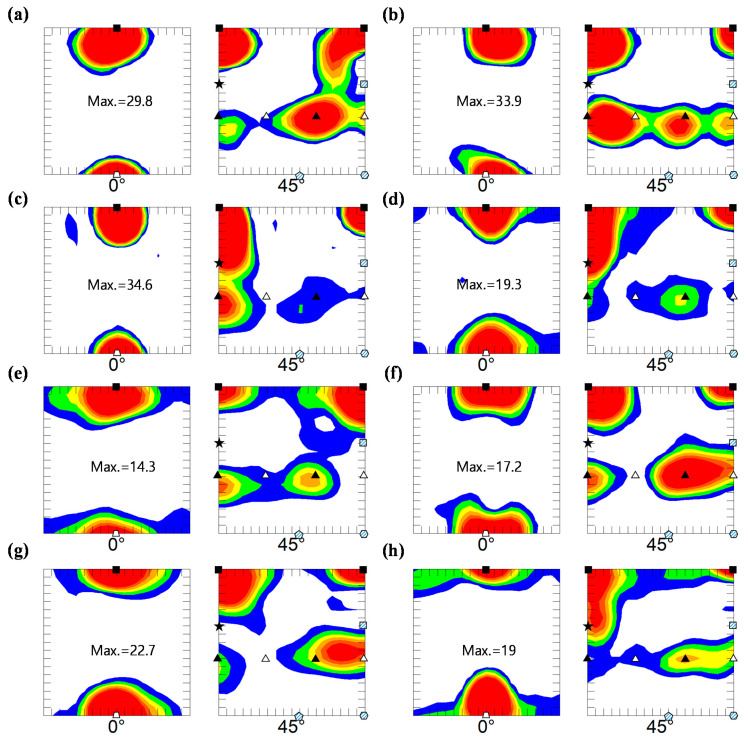
ODF sections with φ_2_ = 0° and 45° of the (**a**–**d**) ESR- and (**e**,**f**) HRDSR-processed samples at different thickness reduction ratios: (**a**,**e**) 20%, (**b**,**f**) 40%, (**c**,**g**) 60% and (**d**,**h**) 80%.

**Figure 9 materials-15-00752-f009:**
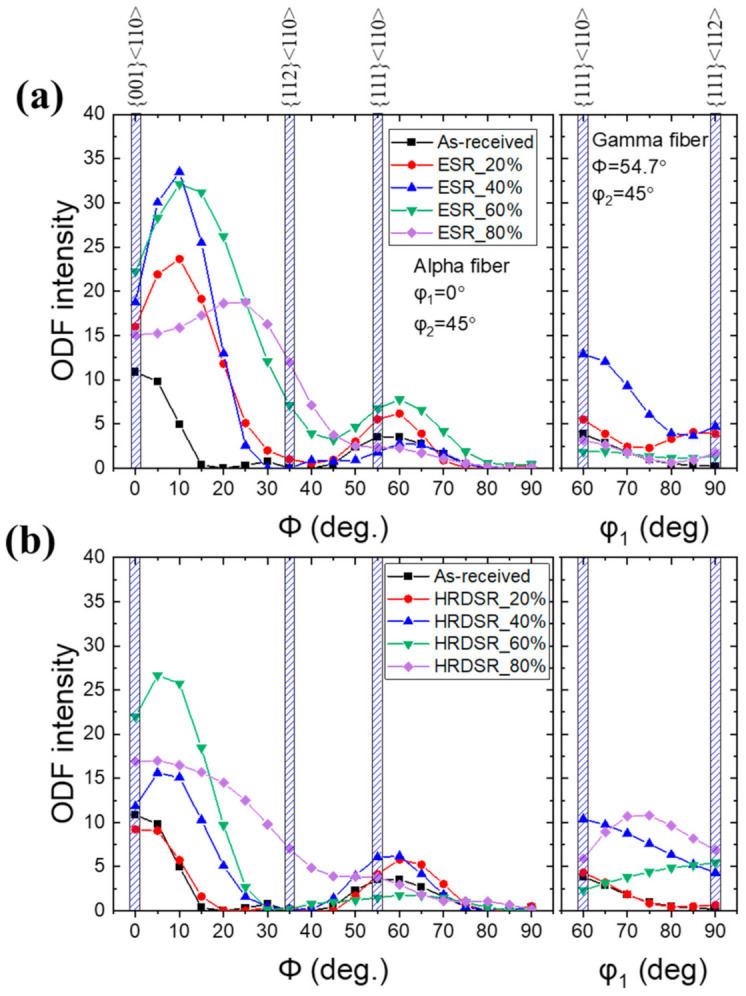
ODF intensities of α- and γ-fibers for the (**a**) ESR- and (**b**) HRDSR-processed samples.

**Figure 10 materials-15-00752-f010:**
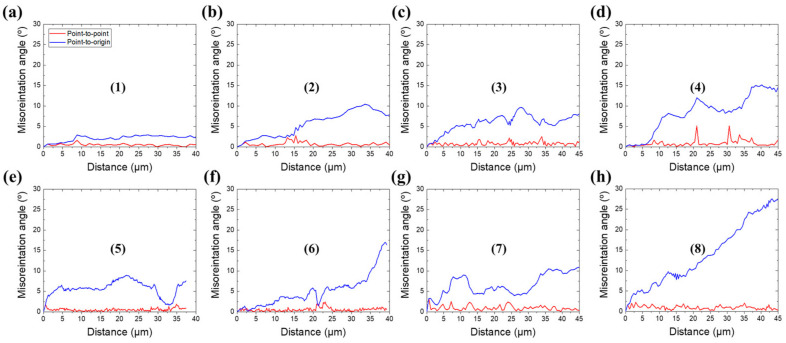
Misorientation profiles marked along the arrows in Figure 4a–h: thickness reductions of (**a**,**b**) 20% and (**c**,**d**) 80% in the ERS-processed materials and thickness reductions of (**e**,**f**) 20% and (**g**,**h**) 80% in the HRDSR-processed materials.

**Figure 11 materials-15-00752-f011:**
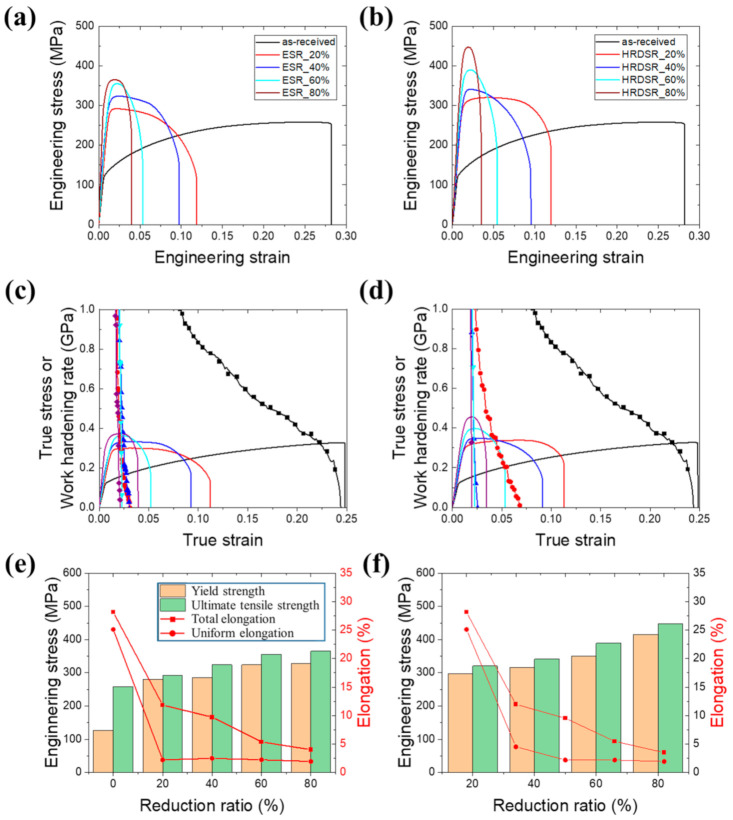
Engineering stress–engineering strain curves of the (**a**) ESR- and (**b**) HRDSR-processed materials at different thickness reduction ratios. Changes in work hardening rate (θ = dσ/dε) and true stress (σ) as a function of true strain (ε) for the (**c**) ESR- and (**d**) HRDSR-processed materials at different thickness reduction ratios. The variation in the yield strength, ultimate tensile strength, and tensile elongation of the (**e**) ESR-processed and (**f**) HRDSR-processed materials at different thickness reduction ratios.

**Figure 12 materials-15-00752-f012:**
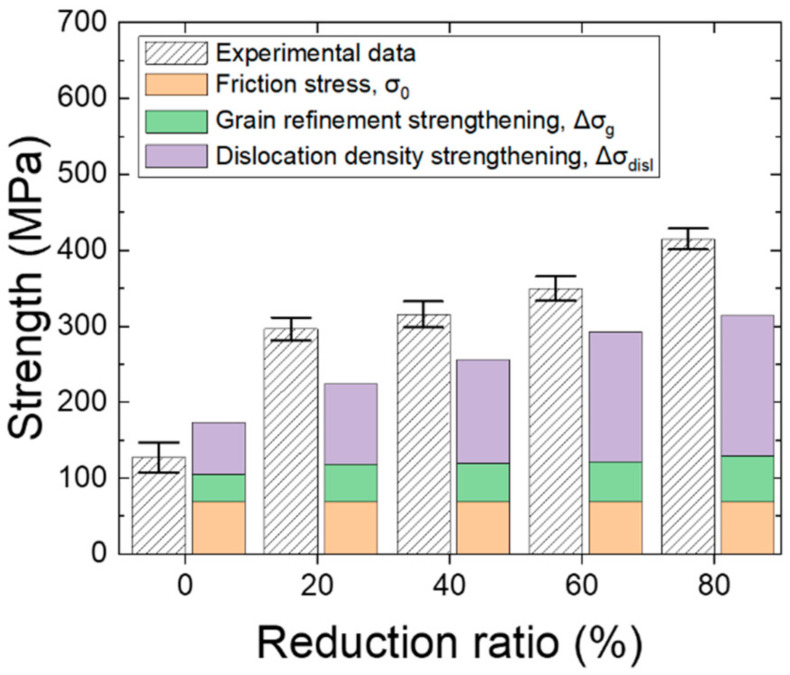
Comparison of the experimental data with the calculated yield strength.

## Data Availability

The raw/processed data required to reproduce these findings cannot be shared at this time as the data also form part of an ongoing study.

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
