# Peer review of "Microstructural and Texture Evolution in Pure Niobium during Severe Plastic Deformation by Differential Speed Rolling"

_materials, 2022, doi:10.3390/ma15030752_

Round 1

Reviewer 1 Report

In this work, the authors studied the evolution of the microstructure, texture and mechanical properties of pure niobium due to equal speed rolling and high-ratio differential rolling at room temperature. The research work is systematic. However, there are also some specific points needed to be addressed.

  1. Page 2, line 65, the unit is lost.
  2. It is better to unify the bars (0%) in Fig. 5 and Fig. 6.
  3. Page 10, line 262. What does σ0 mean?
  4. How does the author calculate the density of dislocation? It is essential to the accurate of the calculated strength.

Author Response

     I appreciate the reviewer's valuable comments.

  1. Page 2, line 65, the unit is lost --unit is added.
  2. It is better to unify the bars (0%) in Fig. 5 and Fig. 6.--bar for 0% is unified.
  3. Page 10, line 262. What does σmean?--This is stated.
  4. How does the author calculate the density of dislocation? It is essential to the accurate of the calculated strength.--Method of calculation of density of dislocations is stated in the experimental method.

Reviewer 2 Report

Article on a high level of metallographic tests performed on modern equipment. Below are some comments that will improve the article's clarity for the reader

  1. The introduction should end with a summary of the literature characterizing the achievements of other researchers and a precise definition of the research problem undertaken by the authors
  2. Authors often use letter abbreviations, not always giving the full name. This makes it difficult to read the article
  3. Please complete the description of the exemplary rolling technology. It is difficult for the reader to find out whether the sample has been rolled once, e.g. up to 40% or in several steps, incl. twenty%; thirty%; up to 40%. I propose an exemplary rolling technology to be presented in the table
  4. Dog-bone shaped tensile samples that's not a fancy term for a research paper
  5. What the authors can say about the post-rolling anisotropy, why samples were taken only along the rolling direction
  6. Was there any skid on the roller with higher speed (HRDSR)?

Author Response

I appreciate the reviewer's valuable comments.

  1. The introduction should end with a summary of the literature characterizing the achievements of other researchers and a precise definition of the research problem undertaken by the authors -- Almost all the research papers on mechanical properties and texture of pure Nb are referred and the important findings are stated in the introduction part.
  2. Authors often use letter abbreviations, not always giving the full name. This makes it difficult to read the article.-- I make sure of giving the full name.
  3. Please complete the description of the exemplary rolling technology. It is difficult for the reader to find out whether the sample has been rolled once, e.g. up to 40% or in several steps, incl. twenty%; thirty%; up to 40%. I propose an exemplary rolling technology to be presented in the table -- A paper about HRDSR technique (a Review paper) is referred and more details on HRDSR equipment and rolling steps (%) are described.
  4. Dog-bone shaped tensile samples that's not a fancy term for a research paper -- This term has been widely used in this field.
  5. What the authors can say about the post-rolling anisotropy, why samples were taken only along the rolling direction-- Due to the shortage of the samples, we could not conduct tensile samples along other directions. However, the currently obtained results are enough to examine the effect of HRDSR on microstructure and mechanical properties of pure Nb and note the difference between HRDSR and ESR. 
  6. Was there any skid on the roller with higher speed (HRDSR)?-- Stated about this in the revised version: No skid occurred during rolling.

Reviewer 3 Report

This article is well written, please add the following information

*Brand from the equipment used in materials and methods.

*More information regarding processing Nb, for example roll diameter, and equipment brand.

*Please explain statistically the differences between the average grain size. 

*Please add the standard deviation in Figure 11 for experimental data.

Author Response

I appreciate the reviewer's valuable comments.

*Brand from the equipment used in materials and methods.

-- More details on HRDSR machine is stated and a proper review paper on this is referred.

*More information regarding processing Nb, for example roll diameter, and equipment brand.
-- More details on HRDSR machine is stated 

*Please explain statistically the differences between the average grain size. 

--Figure 6 caption is revised.

*Please add the standard deviation in Figure 11 for experimental data.

-- Standard deviation is added.

Reviewer 4 Report

1 L62. Please provide some photos of the Nb samples, the experimental process and the deformed samples, like previous research [1-2].

  1. Relationships among the Characteristic Tensile Strain, Curing Age, and Strength of Reactive Powder Concrete. MATERIALS 2021 14 (10) 2660.
  2. Characterization of Loading Rate Effects on the Interactions Between Crack Growth and Inclusions in Cementitious Material. CMC-COMPUTERS MATERIALS & CONTINUA, 2018, 57 (3), 417-446.
  3. L252 Please provide the citation of Eqs.1-3.
  4. L253 Why do you set K=340?
  5. L254 and L261. The bracket is not complete.
  6. l256 Why do you set alpha=0.5 and M=3.3?
  7. Please put the legend in Fig.6a to the top right corner.
  8. The size of the characters in Fig. 11 is so different. Please use the uniform size.

Author Response

I appreciate the reviewer's valuable comments.

1 L62. Please provide some photos of the Nb samples, the experimental process and the deformed samples, like previous research [1-2]. --Figure 3 is newly added.

2. L252 Please provide the citation of Eqs.1-3.-- Proper equations are cited.

3. L253 Why do you set K=340? --More detail on this value and related references are given.

4. L254 and L261. The bracket is not complete.-- completed.

5. Why do you set alpha=0.5 and M=3.3?-- --More detail on this value and related references are given. 

6. Please put the legend in Fig.6a to the top right corner.-- The position of legend is moved.

7. The size of the characters in Fig. 11 is so different. Please use the uniform size.--This is fixed.